# Current State of the Neurotrophin-Based Pharmaceutics in the Treatment of Neurodegenerative Diseases and Neuroinflammation

**DOI:** 10.3390/medsci14010015

**Published:** 2025-12-29

**Authors:** Tatiana A. Fedotcheva, Nikolay L. Shimanovsky

**Affiliations:** Laboratory of Molecular Pharmacology, Pirogov Russian National Research Medical University, 1 Ostrovityanova St., Moscow 117997, Russia; shimannn@yandex.ru

**Keywords:** neurotrophins, neurodegenerative diseases, neuroprotection, neuroinflammation, nerve growth factor, NGF, BDNF, NT-3, NT-4, GDNF, CNTF

## Abstract

Background: The regulation of the synthesis of the nerve growth factor and other neurotrophins is one of the dynamically developing areas of pharmacotherapy of neurological and mental disorders. Despite a large number of studies of various ligands of neurotrophin receptors, only a few have reached clinical application and only for ocular diseases. The aim of this narrative review was to systematize the main progress on neurotrophin-based pharmaceutics; to perform a comparative critical analysis of various therapeutic strategies, elucidate the underlying causes of clinical trial failures, and identify the most promising avenues for future development. Methods: The literature search was conducted in PubMed, Google Scholar, Medline, and EBSCO, and the ClinicalTrials.gov database was used to track current clinical studies, along with the official websites of pharmaceutical companies. The search covered original studies published up to October 2025, with inclusion restricted to articles published in English. Articles describing specific pharmacological compounds that had reached the clinical trial stage were selected. Foundational biological research was referenced to contextually explain the mechanisms of action of the drugs and their therapeutic implications. Results: Recombinant neurotrophins and synthetic molecules, the agonists and antagonists of their receptors, and cell-based gene therapy are promising means for the prevention and rehabilitation of ischemic conditions, as well as the treatment of neuropathic pain and neurodegenerative disorders such as Alzheimer’s disease and Parkinson’s disease. Some of these have undergone clinical trials, yet only neurotrophins for ocular diseases have been implemented in clinical practice: recombinant NGF—cenegermin and recombinant CNTF—Revakinagene taroretcel. The success of these eye drugs is likely attributable to their local administration, improved bioavailability, and low ocular immunoresistance. Conclusions: The study identified limitations and future prospects for neurotrophin-based pharmaceuticals. For future clinical trials, attention should be paid to the pharmacogenetic profiles of the patients and the evaluation of the inflammatory status of the disease. Novel plasma biomarkers of the effectiveness are needed as well as TSPO-PET imaging. Drug delivery systems remain insufficient; therefore, efforts should focus on inducing endogenous neurotrophin production and developing highly selective agonists and antagonists of neurotrophin receptors. It is crucial to establish a favorable premorbid background before neurotrophin therapy to minimize immunoresistance.

## 1. Introduction

Nerve growth factor (NGF), brain-derived neurotrophic factor (BDNF), neurotrophin-3 (NT-3), and neurotrophin-4 (NT-4) are essential for neuronal survival, axon growth, and synaptic plasticity [1]. The regulation of the synthesis and release of neurotrophins is currently being actively studied as it represents a promising strategy in the treatment of pathological conditions such as peripheral neuropathy [2], neurodegenerative diseases [3,4], rehabilitation after strokes [5], and improvement of cognitive functions [6]. Like any other human growth factors, nerve growth factors promote neuronal survival, axon regeneration, and myelin synthesis and improve the synaptic conductivity after neuronal injury [7]. Therefore, searches for the mimetics of nerve growth factors or the agonists of their receptors are being conducted, with the aim to use them in the treatment of neurological and psychiatric disorders.

Since the receptors of nerve growth factors are expressed not only in nerve tissues but also in tumors, the antagonists and blockers of their receptors are being tested as potential antitumor drugs. NGF was the first discovered member of the neurotrophic factor family [8]. Neurotrophins also include BDNF, GDNF (glial cell line-derived neurotrophic factor), NT-3, NT-4, and NT-5, as well as poorly characterized factors such as EPO (erythropoietin), CNTF (ciliary neurotrophic factor), and FGF (factor of fibroblast growth). Neurotrophin-6 (NT-6) and NT-7 have so far been identified only in fish (Xiphophorus and zebrafish *Danio rerio*, respectively) [9].

There is no unified classification of nerve growth factors, nor is there a systematization of the data of research on nerve growth factors and the modulators of their receptors on the basis of which drug candidates have been approved or not. New data on nerve growth factor receptors continue to emerge, and the pharmacological regulation of these receptors immediately becomes the focus of numerous preclinical studies. Unfortunately, until now, most clinical trials involving recombinant nerve growth factors and various activators of their receptors have failed to show success in treating the neurodegenerative diseases such as Alzheimer’s disease (AD), Parkinson’s disease (PD), or multiple sclerosis. Promising efficacy data for LM11A-31, now progressing to Phase III trials for Alzheimer’s disease, has been reported. For many neurotrophin-modulating agents, favorable safety profiles in early-phase studies support the potential for clinically meaningful efficacy in later-stage trials. There are a few FDA-approved drugs for ocular therapy—Oxervate and Revakinagene taroretcel (NT-501), and at the latest phase of clinical trials is Levicept for osteoarthritis treatment.

In this review, we have attempted to consolidate the existing data on neurotrophin-based pharmaceutics and their role in the modulation of neuroinflammation and outline future prospects for their application in the treatment of neurodegenerative diseases. This review summarizes the current landscape of neurotrophin-based drug development, highlighting the promise of the developed compounds for combating neurodegenerative diseases and associated neuroinflammation.

## 2. Classification of Nerve Growth Factors and Their Receptors

The classification of nerve growth factors is contradictory and rather subjective. Newly discovered factors constantly complement the list of NGFs. Currently, there exists no published work that classifies neurotrophins. The unified classification of neurotrophins will aid in optimizing neurotrophin-targeted strategies for treating neurodegenerative and neuroinflammatory diseases. The most common classification includes growth factors that promote differentiation and survival in the nervous system: NGF, BDNF, NT-3, and NT-4 [3].

The neurotrophic factors that exert cytokine-like activity include the GDNF and CNTF. Therefore, all neurotrophins can be divided into three main groups:(1)NGF, BDNF, NT-3, NT-4;(2)Glial cell line-derived neurotrophic factor GDNF;(3)Ciliary neurotrophic factor CNTF

Another classification divides nerve factors into three main families: classical neurotrophins (NGF, BDNF, NT-3, NT-4), neuropoietic cytokines, and glial cell line-derived neurotrophic factor ligands (GFLs). According to this classification, GDNF belongs to GFLs. Among GFLs, GDNF binds to, and activates GFRα2; neurturin (NRTN) binds to, and activates GFRα2; artenine (ARTN) binds to, and activates GFRα3; and persephin (PSPN) binds to, and activates GFRα4 [10]—Figure 1.

GDNF, NRTN, ARTN, and PSPN, through binding with the appropriate GFRs, i.e., GFRα1–GFRα4, respectively, initiate a signaling cascade via the RET (REarranged during Transfection)—the tyrosine kinase receptor that activates ERK/MAPK, PI3K/Akt, and other survival and growth pathways [11].

Classical neurotrophins include NGF, BDNF, NT-3, and NT-4 [12]. GDNF and its structurally related analogues NRTN, ARTN, and PSPN are also neurotrophins obtained from cell lines. CNTF, IL-6, and neuropoietin belong to neuropoietic cytokines. Based on their origin, neurotrophins can be classified as follows:(1)Neurotrophins (NTs);(2)GDNFs and GDNF family ligands (GFLs);(3)Neuropoietic cytokines, also known as the interleukin-6 family (IL-6).

Castilla-Cortázar et al. [10] proposed a different classification system, which accommodates the newly identified, atypical neurotrophin CDNF/MANF (cerebral dopamine neurotrophic factor (CDNF) and the mesencephalic astrocyte-derived neurotrophic factor (MANF). This family includes several protein members that structurally differ from the other neurotrophins and have different mechanisms of action.

According to Castilla-Cortázar [10], neurotrophic proteins can be divided into four groups:(1)NGF, BDNF, NT-3, and NT-4;(2)Neurturin (NRTN), artemin (ARTN), and persephin (PSPN);(3)Cytokines including IL-6, IL-11, IL-27, leukemia inhibitor factor (LIF), ciliary neurotrophic factor (CNTF), cardiotrophin 1 (CT-1), neuropoietin, cardiotrophin-like cytokine (CLC), also known as novel neurotrophin 1 (NNT1), and meteorin;(4)CDNF/MANF.

In the context of drug development, the most widely studied neurotrophins are the members of the first group of any of the above-mentioned classifications, the classical nerve growth factors. Various strategies are aimed at modulating the receptors and signaling, activated by neurotrophins. In particular, NGF, BDNF; and GDNF, CNTF, and NT-3 are being actively developed in clinical and preclinical studies.

The receptors of these neurotrophins are as follows:-High-affinity nerve growth factor receptor TrkA (NTRK1) for NGF;-TrkB (NTRK2) for BDNF and NT-4;-TrkC (NTRK3) for NT-3;-P75NTR (NGFR, LNGFR) for NGF, BDNF, NT-3, and NT-4/5;-CNTFRα, IL-6Rα (CD126), LIFR, p-GP130 for CNTF [13];-GFRα-RET complex for GDNF: GFRα1 for GDNF, GFRα2 for NRTN, GFRα3 for ARTN and GFRα4 for PSPN;-GRP78-NBD (nucleotide-binding domain of GRP78) for both CDNF and MANF-IRE1α (inositol-requiring enzyme 1) for both CDNF and MANF [14].

Recently, Chaldakov G.N. and coauthors introduced a new term, trackins, for Trk-targeting drugs [15]. They suggested that TrkA (tropomyosin receptor kinase A) agonists could be potential drugs for the treatment of neurotrophin deficiency linked to cardiometabolic disorders and neurodegenerative diseases. Trk antagonists, particularly TrkA inhibitors of NGF, are being investigated for prostate and breast cancer, pain, and arrhythmogenic right-ventricular dysplasia [15].

The expression patterns of neurotrophins and their receptors can vary significantly in individuals, which can lead to different outcomes of clinical trials based on neurotrophin pharmaceuticals. The polymorphisms in the genes encoding neurotrophins influence the effectiveness of the treatments as well. Further research is needed to develop drug delivery systems for neurotrophins since they do not penetrate the blood–brain barrier (BBB). Until now, nothing is known about the clinical efficacy of neurotrophins encapsulated in nanoparticles or liposomes: many studies have been conducted only in cell lines and animal models. The most promising strategies are gene- and cell delivery systems [16].

## 3. NGF

NGF was one of the first to be identified by Rita Levi-Montalcini, and it seemed that its clinical use would be beneficial in almost all neurological disorders [17]. There are two main receptors for NGF, TrkA, and LNGFR/p75NTR [18]. NGF/TrkA is one of the few nonopioid targets for the treatment of chronic pain [19,20], but many clinical studies that used recombinant NGF to treat various pathologies accompanied with pain have not been successful. Currently, there is only one FDA-approved NGF-based drug Oxervate for the treatment of neurotrophic keratitis [21].

NGF has a structure similar to insulin-like proteins; it binds to, and activates two receptors: TrkA and the P75 neurotrophic receptor (p75NTR) [22]. The main role of NGF is to support the development and survival of different neuronal populations.

In addition to its well-known neurotrophic function, NGF has a regenerative effect on human skin pressure ulcers, stimulates melanocytes, enhances hair growth and pigmentation [23], and promotes angiogenesis through increases in VEGF and FGF (fibroblasts growth factor) expression in endothelial cells [24]. Eye topical NGF treatment stimulates corneal innervation and healing and increases the viability of corneal stem cells, as well as stromal and endothelial cells [25]. Besides neurons, NGF regulates the viability and proliferation of other target cells, such as mast cells, T and B lymphocytes, granulocytes, monocytes, keratinocytes, endothelial cells, and hormone-secreting cells in the reproductive system [25]. Possibly, there is a biofeedback between NGF and neurosteroids: lowered levels of NGF can stimulate the production of neurosteroids, which subsequently induce NGF synthesis.

NGF is expressed in various organs and tissues: according to the Human Protein Atlas, the highest expression of NGFR is in the heart muscle, ovaries, and the testes [26]. It is known that NGF promotes the release of corticotropins and lactotropins in the pituitary gland and stimulates the production of catecholamines in the adrenal medulla as well as of steroids and spermatogenesis in the testicles; in addition, it enhances the production of steroids and the maturation and ovulation of follicles in the ovaries [25]. The fact that NGF regulates the biosynthesis of sex hormones may indicate that its receptors may be not only TRKA and p75NTR, but also steroid hormone receptors and steroidogenic enzymes, and that the binding of NGF to blood plasma proteins may play a key role in the transport and distribution of sex hormones, influencing its bioavailability.

### 3.1. NFG Agonists

The recombinant human NGF, a modified form of NGF, human beta-nerve growth factor (β-NGF)-(1-118)-peptide (non-covalent dimer) produced in *E. coli* (cenegermin, Oxervate^®^ eye drops), was approved by the FDA in 2018 for treating neurotrophic keratitis. It selectively binds TrkA without activating p75NTR and, therefore, without inducing pain [27]. The dosage regimen is 20 µg/mL solution as one drop into the altered eye 6 times daily for 8 weeks. The most common adverse reaction reported for cenegermin is eye pain, which occurs in approximately 16% of patients [27].

A team of Italian scientists has developed a painless human nerve growth factor (hNGFp), a recombinant mutated form of the neurotrophin NGF. The introduced mutations reduce the binding affinity to p75NTR, resulting in at least a tenfold alleviation in algogenic activity in vivo [28].

Current research on hNGFp in animal models focuses on its neuroprotective potential in glaucoma and optic pathway gliomas [28]. However, until now, no neuroprotective effects in late-stage degeneration were demonstrated in a murine retinitis pigmentosa model either after its intravitreal injection or intranasal delivery [29].

The limited clinical efficacy of NGFs is largely attributed to their molecular structure because they are peptides (large molecules, rapid degradation, poor BBB penetration) [28]. There are also limitations of NGF therapy regarding the profiles of expression and polymorphisms of NGF itself and its receptors (TrkA and NGFR/p75NTR).

Single nucleotide polymorphisms (SNPs) in NGFR/p75NTR are associated with an increased risk of AD [30]. NGF gene polymorphisms correlate with psychiatric addictions [31]. Specific variants of NGF (NGF rs6330, NGFR rs2072446/rs734194) can modulate ischemic stroke susceptibility [32].

Limitations also involve the trophic/angiogenic properties of NGF, which may include the following:Promote tumor progression (e.g., NGF upregulates p75NTR in triple-negative breast cancer);Induce chemotherapy resistance in triple-negative breast cancers by inhibiting apoptosis;Potentially trigger carcinogenesis [33].

Nerve cells interact with the tumor microenvironment in colon, rectal, prostate, head and neck, and breast cancers, and this can lead to tumor progression. Neural invasion stimulates cancer cell dissemination [34]. In this way, NGF can promote neural invasion of cancers [35]. The high density of nerves within tumors is associated with poor clinical outcomes, suggesting that supply with NTs promotes more aggressive tumor growth [36].

Tumor cells usually infiltrate small nerves. In this microenvironment, the cross-talk between nerves and cancer cells, mediated by neurotrophic factors, facilitates tumor invasion and metastasis. Thus, the antagonists of NGF synthesis are promising in the treatment of the tumors with the high neural invasion, such as pancreas, head and neck, and colorectal cancers, but at the same time there is evidence for the possible serious cancerogenic side effects of NGF agonists and mimetics [37]. Future clinical trials should take this problem and other above-mentioned limitations into account.

A more intriguing task is the possible application of NGF agonists and mimetics in the acute period of ischemic stroke and traumatic brain injury (TBI). During TBI, NGF levels increase in response to the damage. This effect makes NGF a promising candidate for therapies targeting both neurological diseases and TBI [38].

### 3.2. NGF Antagonists

Philip A Barker and coauthors analyzed the human diseases in which increased levels of NGF were observed. Among these were rheumatoid arthritis, osteoarthritis (OA), chronic headache, low-back pain, cystitis, and pancreatic cancer [39]. The results demonstrated that NGF expression was increased during inflammation, since the inflammation was a feature inherent in all of the pathologies mentioned above. This explains a high incidence of hyperalgesia, a commonly occurring side effect of NGF in clinical trials. NGF treatment can induce hyperalgesia through rapid (seconds and minutes) and long-term (hours and days) signaling, which involves nociceptive ion channels, receptors, and peptides. However, neither monoclonal antibodies targeting and neutralizing TrkA (e.g., tanezumab and fasinumab) nor small-molecule inhibitors of NGF/pro-NGF that disrupt the binding of NGF/pro-NGF to TrkA and p75NTR (e.g., PD90780, Ro 08-2750) [40], nor small-molecule TrkA inhibitors (such as oral ASP7962 and intra-articular GZ389988A) have demonstrated the superiority in the treatment of OA pain over traditional nonsteroidal anti-inflammatory drugs (NSAIDs) in clinical trials [41].

Tanezumab has been approved by the FDA, but the development program for the treatment of OA pain was discontinued in 2021 because of serious side effects (rapid OA progression due to the blockade of nociceptive signaling, which leads to the overuse of damaged joints). The development of the other anti-NGF antibodies fasinumab and fulranumab as a monotherapy of OA was stopped as well in 2021.

Despite the lack of pronounced clinical benefits, research on fasinumab for managing OA continues. In particular, a combination therapy, such as fasinumab (1 mg every 4 weeks) with diclofenac (75 mg twice daily) or celecoxib (200 mg daily), has shown some promising results. Improvements in the physical function (−0.64 vs. −0.31; nominal * *p* * < 0.05) and pain relief (−0.63 vs. −0.39; * *p* * = NS) were greater with fasinumab compared to NSAIDs alone [42].

The adverse effects observed in clinical trials could result from suboptimal treatment regimens, including inappropriate doses, administration routes, and treatment duration. The multicentral clinical trials of fasinumab and fulranumab were stopped, but the overall data from the network meta-analysis show that monoclonal NGF antibodies provide a significantly greater pain relief and functional improvement in OA compared to NSAIDs and opioids, when all three variables (dose, administration route, and treatment duration) were combined in the analyses [43]. Monoclonal NGF antibodies do not give rise to severe adverse effects if these parameters are properly selected (low dosage, treatment duration of less than three months, and local injections (e.g., intra-articular) reduce systemic exposure). More studies are needed to confirm these findings [44]. Until now, anti-NGF antibody treatment of hip and knee OA is still very expensive, USD 400–600 per dose, which also limits further progress in its implementation.

DS002 is a novel analgesic drug based on the anti-nerve growth factor antibody, independently developed by Guangdong Dashi Pharmaceuticals [45]. It blocks the binding of NGF to the TrkA receptor, making it incapable of stimulating the corresponding sensory neurons, which in final analysis, leads to an analgesic effect. This monoclonal antibody is also being investigated for neuropathic pain, fibromyalgia, and cancer pain. Recently, a Phase I clinical trial of DS002 has been completed, which demonstrated safety within the dose range of 0.5 mg–20.0 mg [45,46].

It has been hypothesized that the pathological (pain-inducing) effects of excess NGF are mediated not through its high-affinity receptor TrkA, but primarily through its low-affinity p75NTR, since NGF also binds to, and activates this receptor [18]. While NGF promotes the survival and regeneration of both motor and sensory neurons, it simultaneously increases nociceptor sensitivity. This effect leads to significant neuropathic and joint pain in clinical trials of recombinant NGF.

Particular attention is now focused on modulating the p75NTR signaling, as this receptor binds to several neurotrophins: proNGF, NGF, BDNF, NT-3, and NT-4/5. It is involved in the pathways that regulate both neuronal survival and cell death.

The search for selective inhibitors of the low-affinity p75NTR resulted in the development of LEVI-04, a p75NTR fusion protein and at the same time a NT-3 inhibitor (trkC receptor antagonist) for the treatment of OA [47].

The small molecules LM11A-31, LM11A-24, EVT901, and THX-B, highly specific p75NTR antagonists, in preclinical studies were shown to reduce neuroinflammation and neuronal death in an animal model of AD and traumatic brain injury [48]. The most developed drug candidates (approaching regulatory approval or being in advanced clinical phases) among the discussed are LEVI-04 and LM11A-31.

LEVI-04 binds excess NGF and has been designed for patients with knee OA [39]. LEVI-04 binds to NT-3 with high affinity and weakly to NGF, targeting the restoration of neurotrophin homeostasis by scavenging excess neurotrophins.

In a randomized controlled Phase II trial, LEVI-04 demonstrated alleviation of pain, impaired function, and other patient-reported outcomes without evidence of increased structural joint pathologies [49].

Another NGFR/p75^NTR^ antagonist is LM11A-31 [18]. This compound was developed for the treatment of AD since Aβ peptides can bind to NGFR/p75^NTR^ and subsequently cause neurotoxicity. LM11A-31 has passed a Phase II clinical trial and is currently underway in patients with mild and moderate AD. The results of Phase II trials demonstrated safety and preliminary efficacy, with the drug being suitable for oral administration. Phase III trials are planned with support from AstraZeneca.

Another class of pharmaceuticals that is worth mentioning includes TRK inhibitors. Their mechanism of action relies on binding to the ATP-binding site within the kinase domain. A crucial point is that this site is only accessible in oncogenic, mutated, or fused TRK proteins, and not in the native, the NGF-activated TrkA receptor. TRK inhibitors do not bind native TrkA activated by NGF. There are two drugs: larotrectinib (LOXO-101) and entrectinib; both are indicated for the treatment of advanced or metastatic solid tumor cancers with neurotrophic tyrosine receptor kinase (NTRK) gene fusions. TRK kinase inhibitors (larotrectinib and entrectinhib) have passed several clinical trials and have been approved by the FDA [50]. These tumor-agnostic drugs are modern pharmaceuticals for precision oncology and show high effectiveness in the treatment of NTRK fusion-positive cancers [50].

The diversity of receptors for nerve growth factors and the ability of the same growth factors to bind to different receptors led to the development of three of the most promising therapeutic agents to date: agonists and antagonists of NGF receptors. The TrkA agonist cenegermin (Oxervate, eye drops) was approved by the FDA in 2018 for the treatment of neurotrophic keratitis. The p75NTR antagonists are Levi-04 and LM11A-31.

LEVI-04 is a fully human chimeric fusion protein consisting of the Fc fragment of human immunoglobulin G1 fused to the extracellular domain of p75NTR. In addition to the blockade of NGF, LEVI-04 provides analgesia by inhibiting the activity of NT-3, thereby modulating the elevated neurotrophin levels associated with osteoarthritis.

A Phase II randomized, double-blind trial involving 518 participants demonstrated significant analgesic effects in all dose groups. The mean reduction in WOMAC pain score from baseline exceeded 50% for all three doses of LEVI-04, and all measured parameters were statistically significant compared to the placebo (ClinicalTrials.gov Identifier: NCT05618782) [49].

## 4. BDNF

BDNF, like NGF, stimulates neurogenesis, synaptic plasticity, and neuronal survival. For BDNF, the balance between its precursor form and the mature form is important for the normal functioning of neurons. The decreased synthesis and release of BDNF are strongly implicated in the pathophysiology of neurodegenerative diseases such as AD, PD, multiple sclerosis, and amyotrophic lateral sclerosis [51]. Different therapeutic strategies are explored to activate BDNF signaling. TrkB is a specific receptor for BDNF. BDNF activators, as for instance, TrkB agonists and TrkB mimetics, can restore the synaptic function and ameliorate cognitive deficits in neurodegenerative diseases [52].

BDNF also has protective and neuroregenerative effects, particularly, in the hippocampus during blast traumatic brain injury (bTBI), as recently shown in a rat model [53]. However, the application of BDNF/TrkB receptor agonists and BDNF mimetics can cause different side effects. Like immune cytokines, BDNF plays a dual role in inflammation: it can be protective in the acute phase and harmful during chronic inflammation. Immune cells, in particular, T cells and macrophages, produce BDNF during allergic asthma, which leads to the increased release of neurotransmitters and neuronal muscle contraction. Microglia secrete BDNF in response to TNF-α and IL-6 [54]. Increased BDNF levels promote microglial proliferation and activation during immune reactions [55]. Extensive production of BDNF is linked with the development of epilepsy by inducing neuronal hyper-excitability; the BDNF serum level correlates with the severity of epileptic seizures [56,57].

On the contrary, a recent Indian clinical study on 74 adult (18–50 age) persons with epilepsy showed that there is a different relationship between MoCA scores and the level of BDNF. It was revealed that serum BDNF levels were reduced in the group of patients with cognitive impairments [58]. Therefore, higher levels of BDNF are also unfavorable: changes in the individual BDNF expression may be a helpful diagnostic tool for assessing the impairment of the cognitive functions and mood disorders as well as a marker of the effectiveness of pharmacological treatments.

### 4.1. BDNF Agonists

The known direct agonists of the BDNF receptor TrkB are the natural compound 7,8-Dihydroxyflavone (7,8-DHF) [59] and the small-molecule partial agonist LM22A-4 [60]. There are compounds that are positive allosteric regulators of TrkB, for example, ACD856.

7,8-DHF crosses the BBB and mimics the neurotrophic effects of BDNF and NT-4 [61]. In animal models in more than 150 studies, it was shown to be effective in the treatment of the damaged brain. Supplementation with 7,8-DHF may serve as therapy or, at least, as an adjuvant for the treatment of pathologies affecting not only the brain but also body functioning [62]. Recently this molecule has been shown to modulate aggressive behavior in a fish model [63].

No human studies have been conducted with 7,8-DHF or LM22A-4. In a mouse model of pediatric traumatic brain injury, LM22A-4 showed neuroprotection and remyelination in the perilesional external capsule, reduced reactive gliosis, and the loss of tissue volume [64]. In an animal model of Rett syndrome, LM22A-4 improved hippocampal-dependent object location memory and restored hippocampal long-term potentiation [65]. In rats, LM22A-4 induced vascular relaxation with the same efficacy as acetylcholine [66]. In a mouse model of spinal cord trauma, LM22A-4 reduced the histologically measured degree of tissue injury and significantly restored the limb function [67].

Since LM22A-4 poorly penetrates the BBB when administered systemically, animal studies are ongoing with improved formulations to enhance bioavailability, for example, with LM22A-4-loaded smart mesoporous balls (LM22A-4-SMB-3). Thus, it was found that LM22A-4-SMB-3 improves post-stroke recovery in a mouse model of ischemic stroke, reduces apoptosis and glial activation, increases the phosphorylation of TrkB and Akt, boosts neurogenesis, and diminishes brain atrophy after the stroke [68]. This molecule has not yet been approved for clinical trials.

ACD 856 has another mechanism of action: it acts through the allosteric activation of the TrkB receptor, which results in the stimulation of the neurotrophin signaling pathways. This molecule showed precognitive and antidepressant efficacy in various animal models [69]. Currently, ACD 856 is under investigation in clinical trials as an oral drug for the treatment of AD and other disorders with declined cognitive functions, as well as depression and neuropsychiatric diseases. Recently, a clinical trial with an oral dose of 1–150 mg was conducted, including 56 healthy subjects, which demonstrated a safe and well-tolerated profile of ACD 856. ACD856 was shown to pass the BBB and to induce dose-dependent treatment-related changes in the qEEG parameters, indicating the central mechanism of action [70,71].

The neurosteroids DHEA, progesterone, allopregnenolone [72], ACTH analogs, as well as the antidepressants amitriptyline and esketamine [73] have the ability to activate the release of BDNF. AAV-based BDNF gene therapy for AD models in mice [74] and cell therapy [75] are being actively developed in this field of research [76].

In addition to pharmacological regulation, the synthesis of endogenous BDNF can be stimulated by physical exercises [77,78]; however, not all individuals show a direct relationship between the physical exercise and BDNF release. There is a frequently occurring polymorphism in relation to BDNF, which affects the predisposition to neurodegenerative diseases and the lack of the effect from aerobic physical exercise. In some European populations, the frequency of occurrence of BDNF Val66Met single-nucleotide polymorphism can reach up to 80% [79]. This polymorphism is a modern task of neurobiological investigations, as it has a strong impact on the development of different neurodegenerative and psychiatric disorders. The BDNF Val66Met is a common single-nucleotide polymorphism (SNP rs6265) associated with reduced release of BDNF [80].

The most important limitation of the clinical application of NGF and BDNF is neuropathic pain, which can occur due to signaling via the low-affinity pan neurotrophin receptor at 75 kDa (p75NTR), since both of these growth factors bind to, and activate this receptor [81]. Therefore, selective TrkB agonists are preferred for the development of neurotrophic and neuroprotective drugs.

### 4.2. BDNF Antagonists

BDNF blockade has no therapeutic application because this neurotrophin is essential for the neuronal viability and function. For research purposes, compounds such as ANA-12 (a selective TrkB receptor antagonist) and K252a or AG879 (less specific tyrosine kinase inhibitors) are employed. Monoclonal antibodies against TrkB or TrkB-Fc (TrkB-IgG) fusion proteins are used in experimental settings, such as in vitro studies and animal research.

## 5. NT-3

NT-3 is structurally related to other neurotrophins like BDNF [82]. NT-3 stimulates the embryonic development of the derivative of the neural crest and the central nervous system and also induces the axon growth and repair [83,84].

NT-3 preferentially binds to TrkC, although it can also realize signaling through TrkA and TrkB [85]. NT-3 promotes the myelination and regeneration of peripheral nerves after chronic denervation via the TrkC/ERK/c-Jun pathway [86]. NT-3 specifically modulates immune response since TrkC is expressed by Th2 cells; the Th1/Th2 balance is regulated through the interaction of NT-3 with TrkC [87]. The NT-3/TrkC system is responsible for the maintenance of Th2-dependent immunity [88]. Due to the immunomodulatory and anti-inflammatory properties of NT-3, and its possibility to induce regeneration and myelination, currently NT-3- based AAV1-gene therapy of multiple sclerosis is under investigation in a mice model [88].

Until now, recombinant NT-3 has not been tested in any clinical trials and has not yet been approved for humans [86], but the studies are continuing in animal models. Recently, a novel intrathecal administration method using an in situ hyaluronic acid-modified heparin-poloxamer hydrogel loaded with NT-3 for the direct delivery of NT-3 to the spinal cord of rats with spinal cord injury has been tested. This gel was retained alongside the cord for at least one week and was safe [89].

The level of NT-3 is decreased in depressed patients of all severity degrees and increased in both manic and depressed bipolar patients. NT-3 may exert antidepressant actions, since NTRK3 polymorphism may be involved in the pathophysiology of panic disorder. Therefore, the polymorphisms of NTRK3 and miRNAs posttranscriptional regulation should be considered in future clinical trials [90].

The polymorphism of the *trkC* and its posttranslational modification significantly influence the NT-3 signaling. It was shown that *trkC* mRNA can be spliced to the truncated isoform TrkC.T1. In this case, NT-3 causes an increased production of TNF-α, which leads to inflammation and neurotoxicity [91].

Less is known about the specific action of NT-4/5 since it binds to ands predominantly activates the TrkB receptor. This receptor is also a specific target for BDNF; therefore, until now, more advanced animal and human studies have been related to BDNF-modulating strategies rather than NT-4-modulation. In vivo studies revealed that NT-4/5 attenuated the loss of motor neurons and, similarly to BDNF, reduced neuronal inflammation [92].

In rat cortical neurons, NT-4 caused a more prolonged activation of TrkB than BDNF and reduced TrkB ubiquitination [93]. The molecular mechanism by which TrkB activation by BDNF and NT-4 leads to diverse outcomes is unknown [93].

The goals of further studies along this line will be to determine the NT-3 levels in Charcot-Marie-Tooth disease, peripheral neuropathy, and chronic inflammatory demyelinating polyneuropathy and to elucidate whether it is possible to restore the lowered levels of NT-3 in the serum using the gene therapy [94,95,96]. The validity of this research approach would be strengthened by the demonstration of a robust correlation between serum NT-3 concentrations and the corresponding neuropathology.

### 5.1. NT-3 Agonists

The agonists of NT-3 and NT-4 are primarily recombinant NT-3 and NT-4 proteins, which have not yet been widely investigated; there are no available results from either animal or human trials. Small molecules acting as TrkB agonists (for example, 7,8-DHF) may also serve as functional agonists for these neurotrophins.

The delivery of full-length NT-3 to the CNS after systemic administration is limited; therefore, different low-molecular mimetics such as dipeptides GTS-301 and GTS-302 have been recently synthesized and explored in models of rats with opioid dependence, in particular, morphine withdrawal symptoms. These mimetics reduced the specific morphine withdrawal symptoms in a dose-dependent manner and activated TrkC and TrkB receptors after intraperitoneal injection in mice [97].

Due to cyclization, the cyclic dipeptide analogue of NT-3, c-[YAEHKSHRGEYSV], exhibits, along with enhanced stability and bioavailability, increased selectivity for TrkC. This dipeptide stimulated BDNF and VEGF expression as well as VEGF release and induced the expression of Trks and VEGFRs in SH-SY5Y cells [98]. Due to its ability to differentiate into neurons upon neurotrophin addition, this cell line represents a valuable in vitro model for studying neurodegenerative disorders [99].

### 5.2. NT-3 Antagonists

The antagonists of NT-3 and NT-4 include antibodies (NT-3 and NT-4/5 monoclonal antibodies), peptidomimetics based on the β-turn structure of NT-3, which blocks ligand-independent TrkC activation [91], and inhibitors of TrkB/C (GNF-5837 is selective inhibitor of TrkB/TrkC tyrosine kinases) [100].

K252a can also be classified as an antagonist of NT-3. K252a is a chemical compound that acts as a blocker of Trk receptors, which are involved in the biological effects of BDNF. At high concentrations, K252a inhibits the survival-promoting and neurotrophic effects of NTs, but at low concentrations, it can promote survival and differentiation similar to the effects of NTs [101].

Since K252a is a staurosporine analog, the systemic use of this molecule has been restricted due to its toxicity. Pincelli C. and coauthors developed a topical preparation of K252a for the treatment of psoriasis [102,103]. Pharmaceutical companies are also in a search for an anti-NGF therapy for psoriasis, autoimmune arthritis, and pain control.

Recently, larotrectinib has been approved, which has been mentioned before as an NGF antagonist. It also antagonizes the action of NT-3 and BDNF/NT-3 heterodimer. Several studies of this drug as an anti-tumor agent in various cancers have been performed, and patient recruitment is currently underway for a clinical trial entitled “A Pilot Study of Larotrectinib for Newly-Diagnosed High-Grade Glioma with NTRK Fusion”, ClinicalTrials.gov ID NCT04655404. Larotrectinib was fully approved in 2025 as a kinase inhibitor used to treat solid tumors with NTRK fusion [104].

## 6. GDNF

Signaling by GDNF is mediated through binding to a membrane-associated receptor complex consisting of two subunits: a glycosylphosphatidylinositol (GPI)-anchored protein called GFRα (GDNF family receptor α) and the receptor tyrosine kinase RET. The GFRα activates RET, which subsequently initiates intracellular tyrosine kinase (TK) signaling [105].

GDNF was discovered as a secretion product from the rat B49 glioma cell line in 1993 and was suggested to be the innovative therapy for Parkinson’s symptoms since it protected the midbrain dopaminergic neurons from oxidative stress and other injuries and controlled microglial reactivity [106].

RET is critical in the development and function of the nervous system, the male reproductive system, and the renal system. RET signaling is activated by either GDNF ligands or growth differentiation factor 15 (GDF15) via GFRα or GDNF family receptor α-like (GFRAL) co-receptors, respectively [107]. The importance of RET signaling in nervous system development was proven since mutations in this receptor can cause Hirschsprung’s disease, a rare intestinal motility disorder characterized by aganglionic megacolon [107]. RET is also essential for kidney and spermatogonia development. GDNF plays a protective role in rat embryos during chronic hypoxia [108].

GDNF promotes the survival and axonal regeneration of motor neurons and retinal ganglion cells [109] and is a survival factor for various neuronal cell types [110].

The main therapeutic goal of GDNF-targeted approaches encompasses both neuroprotection in neurodegenerative disorders (e.g., PD) and the management of neuropathic conditions, in particular, sciatica and neuropathic pain of diverse etiologies.

Recent findings suggest that GDNF exerts its neuroprotective effect on dopamine neurons through the modulation of microglial activation, since the endogenous levels of GDNF are insufficient to protect dopamine neurons against an inflammatory insult [106]. Therefore, finding the optimal level of GDNF expression is crucial for each person. A significant limitation of the application of GDNF is that its overexpression is also detrimental, as it can lead to hyperdopaminergia, which may trigger the exacerbation of schizophrenia and the onset of psychosis.

In murine models of schizophrenia, GDNF was shown to activate adenosine A2a receptor (A_2A_R), a G protein-coupled receptor that modulates dopaminergic signaling and could be a trigger for hyperdopaminergia. In human cerebrospinal fluid obtained from 69 first-episode psychosis subjects with schizophrenia, the level of GDNF was twofold elevated compared to 44 healthy controls [111].

Exogenous GDNF and constitutive RET activation can cause behavioral hyperactivity. A twofold increase in GDNF expression improves motor ability in mice; this rise is safe and well tolerated, whereas a 3- to 12-fold increase in GDNF initiates hyperdopaminergic side effects [112]. GDNF can exhibit opposing effects (stimulatory or inhibitory for dopamine regulation) in the A9 dopamine cell group versus the A10 dopamine cell group [112].

Poor GDNF diffusion and immune responses generating neutralizing antibodies may contribute to the largely unsuccessful clinical trial results. The challenge of achieving effective brain delivery remains a key obstacle, driving current research focused on GDNF gene therapy approaches. Unaltered GDNF/GFRα1/RET signaling is important for neuroprotection. The development of RET agonists with the possibility to cross the BBB is one of additional strategies to treat neurodegenerative disorders [81].

Drug delivery systems are developing for the efficient supply of GDNF to the brain, since the BBB limits the access to therapeutic agents. Direct infusion methods have shown limited success. Emerging strategies such as gene therapy that use viral vectors and encapsulated cell bio delivery systems offer promising alternatives.

### 6.1. GDNF Agonists

GDNF agonists or mimetics are highly valuable and critically important. Similar to most growth factors, GDNF cannot pass through the BBB. Consequently, the delivery of the gene has emerged as the most widely explored strategy for targeted administration to the brain. Pharmaceutical companies are developing different strategies to deliver GDNF directly to the brain, particularly, in PD, since GDNF is responsible for the motor neuron protection. Continuous infusion of GDNF to the brain through an implanted cannula is also one of the methods that has been clinically investigated. GDNF-based gene therapy for PD is more favorable due to the less invasive procedure, although all the methods using treatment with GDNF did not achieve its primary end point [105].

Research on AAV2-GDNF using animal models was first launched in 1997 and remains ongoing [113]. A Phase II randomized, double-blind, sham surgery-controlled study of the efficacy and safety of intraputaminal AAV2-GDNF in the treatment of adults with moderate PD (REGENERATE-PD) is currently enrolling participants in the USA [114].

Recently, a clinical trial with AAV2-GDNF bilateral intraputaminal infusion in participants with advanced PD achieved 26% mean putaminal coverage and stable motor features with no unexpected adverse events over 60 months [115].

A clinical trial with CNS10-NPC-GDNF transplanted unilaterally into the lumbar spinal cord of 18 ALS (Amyotrophic Lateral Sclerosis) participants in a Phase I/IIa study showed safety during the first year of examination. A positive result of the trial was that the postmortem tissue analysis of 13 participants who died of disease progression revealed at least graft survival and GDNF production [116].

Clinical studies with AAV2-NRTN showed no superiority to sham effectiveness, but revealed safety and longer survival times of patients receiving the therapy; postmortem analysis demonstrated the expression of NRTN both in the putamen and the substantia nigra [117,118]. Since CERE120 was not superior to sham in a randomized double-blind clinical trial, subsequent research was limited.

Next-generation gene therapies for the GDNF delivery are developing to achieve desirable effectiveness. The main positive results of the previous trials pertained to safety, not efficacy. AAV2-GDNF and AAV2-NRTN (Neurturin, CERE-120) for PD [118], CNS10-NPX-GDNF for ALS, all showed safety and were well tolerated during the first phase of clinical trials [116,119].

Clinical trials of GDNF analogs have also been conducted for the treatment of neuropathic pain. The recombinant artemin-like substance neublastin (BG00010, a selective ligand for the GFRα3coreceptor, Biogen Idec Inc., Cambridge, MA, USA), when administrated intravenously, showed efficacy in alleviating the neuropathic pain in Phase I and II clinical trials, but its effect was not dose-dependent. In patients with chronic painful lumbosacral radiculopathy, neublastin improved the average general pain intensity scores at 1-, 3-, and 5-week post-dosing compared to placebo [120]. Unfortunately, neublastin was ineffective in reducing the Likert pain scores [121].

Other methods of delivering GDNF (intravenous infusion or intranasal delivery) have been less developed than the gene therapy. Until now, all these methods do not have superior effectiveness in clinical trials compared to historically used drugs and methods such as DuoDopa pumps, apomorphine pumps, deep brain stimulation, and pallidotomy [122]. Recent advances showed that CDNF is more specific for the treatment of PD; the Phase I monthly intraputamenal CDNF infusions in patients with PD using a bone-anchored transcutaneous port connected to four catheters demonstrated its safety and a favorable tolerability profile [123].

As of 2025, no GDNF-based therapeutics have received the approval of the FDA. Research efforts are currently focused not only on GDNF as a motor neuron protectant but also on diverse ligands targeting its receptors, GFRα1 and RET [124].

BT13, a small-molecule RET receptor agonist, has been evaluated for PD therapy. In addition, it is explored for glaucoma treatment due to its ability to prevent retinal ganglion cell death. In rat studies, BT13 reduced mechanical hypersensitivity and restored the normal expression levels of sensory neuron markers in dorsal root ganglia [125]. It was shown to enhance the dopamine release in animal models [126,127]. Clinical trials of BT13 have not yet been conducted [128].

BT18, an agonist of GFRα1 and RET, is also known as a GDNF mimetic; among other GDNF mimetics are GFRα-1 agonists aminoquinols [129]. NGF also can act as a GDNF mimetic by activating RET independently of GFLs and GFRα receptors [130].

Not only are pharmacological strategies being tested. Voluntary running has been shown in rats to increase the number of cells and motor neuron cell body mass by almost two times, both in young and old animals [131].

### 6.2. GDNF Antagonists

GDNF antagonists are not required in medicine. The scientific efforts are aimed, on the contrary, at increasing the synthesis and release of GDNF. There are selective RET inhibitors, selpercatinib and pralsetinib, that have been approved for the treatment of thyroid and lung cancers harboring RET mutations [132].

## 7. CNTF

Natural and synthetic modulators of the release of NGF, BDNF, GDNF, and NT-3 have been well studied, whereas less is known about the regulation of the biosynthesis and release of CNTF. The CNTF belongs to the family of cytokines that includes leukemia inhibitory factor (LIF), IL-6, IL-11, and IL-17 and is produced primarily by glial cells of the peripheral nervous system: astrocytes, Schwann cells, and microglia, as well as neurons and skeletal muscle cells [133].

The molecular target of CNTF is a receptor complex composed of the CNTF α-receptor (CNTFRα), glycoprotein 130 (GP130), and leukemia inhibitory factor receptor (LIFR). The assembly of the CNTF-receptor complex triggers the activation of multiple intracellular signaling pathways, such as Ras/MAPK [134]. CNTF binds to the cytokine-binding domain of CNTFRa; then, the CNTF-CNTFRa complex interacts with GP130 and the LIFRb immunoglobulin domain that activates signaling chains through the Jak-STAT pathway [135].

Unlike many other neurotrophic factors, the induction of CNTF expression is not associated with the injury; it is expressed constitutively in unaltered peripheral nerves, but is suppressed during neuronal regeneration. A rapid release of CNTF during traumatic injury of Schwann cells is a stimulus for the synthesis of other trophic factors [136].

Recently, the association between the CNTF level in cerebrospinal fluid and AD progression has been found: a higher level of CNTF is related to a slower rate of cognitive impairment [137].

The clinical significance of CNTF has been discovered in relation to ocular diseases such as retinal keratitis, glaucoma, and macular telangiectasia.

In a mice model of degenerating retinas, CNTF showed anabolic and antioxidant activities (restores the glutathione level by the promotion of aerobic glycolysis). These findings opened up a new field of using CNTF in the treatment of retinal degeneration [138]. A CNTF-based drug has been already approved for humans, and the research of its application in protecting photoreceptor cells and slowing vision loss is being continued [139].

### 7.1. CNTF Agonists

Dapiclermin (Axokine) is a recombinant ciliary neurotrophic factor, which was developed in 1990 to treat ALS. It was not effective for ALS, but protective against obesity and insulin resistance; clinical development was halted by the emergence of CNTF antibodies [140]. The development of AAV2-CNTF (NSR-CNTF) gene therapy for retinitis pigmentosa was also discontinued prior to Phase 3 trials because it failed to demonstrate sufficient therapeutic efficacy in earlier clinical stages.

In 2011, a recombinant CNTF implant for local use, NT-501, was explored to slow down the progression of vision loss in geographic atrophy and dry age-related macular degeneration (AMD) [141,142]. Later, this compound was approved as Revakinagene taroretcel (revakinagene taroretcel-lwey; ENCELTO™). The drug is an encapsulated cell-based gene therapy containing 200,000–440,000 allogeneic retinal pigment epithelial cells expressing recombinant CNTF (rhCNTF). Available as a single-dose intravitreal implant, it was developed by Neurotech Pharmaceuticals, Inc. for the treatment of chronic retinal diseases. In March 2025, revakinagene taroretcel received its first approval for the treatment of adults with idiopathic macular telangiectasia (MacTel) type 2 in the USA [143]. The drug shows considerable promise in the treatment of glaucoma, retinitis pigmentosa (RP), and age-related macular degeneration [144].

### 7.2. CNTF Antagonists

CNTF antagonists are not being developed since there is no need for them in the treatment of neurodegenerative diseases and neuroinflammation. The LIFR subunit of the CNTF receptor complex, when activated, can induce undesirable oncogenic signaling. In this context, two LIFR antagonists are under development to treat cancers: the humanized anti-LIF antibody MSC-1 and the LIFR inhibitor EC359. EC359 was also shown to block the signaling mediated by CNTF and LIF [145].

The main functions of neurotrophins are listed in Table 1. The main advances and the neurotrophin-based pharmaceuticals for the treatment of neurodegenerative diseases and neuroinflammation are listed in Table 2.

Table 2 demonstrates that selective small molecules such as LM11A-31, capable of penetrating the BBB, and AAV gene therapy are the most promising strategies.

## 8. Limitations and Future Prospects for Neurotrophin-Based Pharmaceutics

### 8.1. Limitations

An analysis of data on ongoing preclinical and clinical studies of neurotrophin-based pharmaceuticals allows for the identification of several key factors leading to ambiguous and often unsatisfactory results. The reasons for the lack of success and the premature termination of clinical trials for neurotrophin-based drugs can be categorized into four groups.

Firstly, target-related issues stem from the differences between activating a specific receptor subtype versus stimulating the entire neurotrophin molecule, reflecting insufficient target selectivity and polymorphisms in the target gene.

Secondly, resistance-related issues involve the rapid inactivation of neurotrophins and their ligands, alongside the development of immune-mediated resistance and resistant mutations.

Thirdly, delivery-related challenges are due to poor penetration, often confined to a narrow local area near the injection site, and the emergence of various forms of resistance.

Fourthly, disease-stage-related problems occur when neurotrophins or their mimetics are administered too late, after a phenotype of persistent chronic inflammation and a damaged microenvironment has already been established. Under these conditions, neurotrophins cannot effectively reach the injury site or bind to their receptors.

Every therapeutic strategy has its limitations. Recombinant neurotrophins lack selectivity for specific receptor subtypes, cannot cross the BBB, fail to bind to activated microglia, and are immunogenic. Monoclonal antibodies suffer from limited selectivity and organotropism, poor BBB penetration, and the potential for and possible emergence of systemic side effects and immunogenicity. Small molecules are more favorable due to their ability to cross the BBB and high selectivity, but they also face challenges such as systemic action and the development of resistance.

Gene therapy requires improvement in areas such as enhancing vector tropism and preventing neuronal immune activation and immune resistance, which arise from prolonged transgene expression and cause inflammation at the injection site. Suboptimal tropism, mainly for glial cells, and uneven distribution gradients, restricts viral vectors to areas near the cannula, resulting in local neurotrophin expression that fails to reach the damaged brain regions.

All these limitations are interconnected through issues of insufficient selectivity, various resistance mechanisms, suboptimal delivery and poor penetration, and the advanced stage of neurodegenerative disease. Consequently, a more detailed analysis is essential to optimize both preclinical and clinical research in this field.

#### 8.1.1. Selectivity

The difference between receptor subtype activation and pan-neurotrophin stimulation is one of the main reasons for treatment failure. The dual action of neurotrophins—acting as trophins and inducing proliferative cascades and angiogenesis—can be beneficial for neuronal and axonal survival but also poses a risk as a potential cause of tumor growth and proliferation. Non-selective pan-activation of both receptor subtypes, TRKs and p75NTR, may promote invasion, angiogenesis, and metastasis in neuroblastoma, prostate cancer, and breast cancer via TrkA and p75NTR receptors. Neurotrophic TRK genes are recognized as oncogenic drivers for various tumor types [175]. This is precisely why, only in cases of NTRK fusions (accounting for approximately 0.2% of all tumors), monoclonal antibody, TRK inhibitors such as Entrectinib and Larotrectinib are highly effective.

All the mechanisms by which neurotrophins promote neuronal survival and neuroplasticity can also serve as triggers for tumor cell growth and immune tolerance and intensify innervation and blood supply to tumor foci. Through a feedback mechanism, exogenous neurotrophins may inhibit endogenous neurotrophin production and contribute to receptor desensitization.

Pan-activation of neurotrophin receptors affects not only neuronal cascades directly but also non-neuronal mechanisms within the tumor microenvironment (TME). Increased NGF levels can contribute to the establishment of an immunosuppressive TME and induce resistance to immunotherapy [176]. Cancer-associated fibroblasts (CAFs), tumor-associated endothelial cells (TAE), and tumor-infiltrating lymphocytes (TILs) play a central role in tumor progression [177,178]. Signaling through p75NTR on immune cells can suppress the antitumor activity of TILs. CAFs themselves can produce neurotrophins (NGF, BDNF), which, acting via the TrkB receptor on endothelial cells, stimulate neoangiogenesis [177]. This creates a vicious cycle.

A similar feedback loop forms not only during tumor growth but also during neuroinflammation. During inflammation, mast cells release the NGF in high concentrations, leading to the intensive growth of axons and causing elevated pain reception. Elevated NGFs were found in the blood and tissues of patients with autoimmune diseases, including multiple sclerosis, systemic lupus erythematosus, autoimmune thyroiditis, and chronic arthritis [179,180]. At the same time, the NGF activates proliferative cascades to promote tissue repair during the inflammatory response [181]. This creates a vicious cycle in which neurotrophins, particularly NGFs, play a dual role, similar to pro-inflammatory cytokines: they combat inflammation and pain, but their excessive production again leads to chronic inflammation.

Achieving the absolute selectivity is the most promising strategy. Selectivity can be achieved through the computer modeling of specific ligands for the receptor’s extracellular domain or allosteric modulators. For example, LM22A-4 and 7,8-DHF bind to an allosteric site on the extracellular domain of the TrkB receptor and act as “signal enhancers” for endogenous BDNF [177,182].

Therefore, the selectivity of agonists and antagonists in neurotrophin-based pharmaceuticals is particularly crucial and pertains to NGF- and BDNF-based strategies. This is because TrkA and TrkB, to which these neurotrophins bind selectively, initiate numerous signaling cascades, while the low-affinity p75NTR appears to modulate their activity and may, in fact, regulate their excessive activation. The precise influence of the low-affinity receptor p75NTR on Trk signaling is still not fully understood. p75NTR has a similar affinity for each of the neurotrophins, but NGF treatment induces neuronal survival signaling, whereas BDNF treatment can mediate apoptotic signaling [183].

Regarding anti-NGF strategies, the blockade of p75NTR is preferred, because this low-affinity receptor is particularly active during inflammation. It binds to the immature proNGF and, through the p75NTR/sortilin complex, induces neuronal death, chronic pain, and neuroinflammation [184].

Increasing the level of mature neurotrophins is one of the strategies to enhance therapeutic efficacy. In neurodegenerative diseases, the expression of p75NTR (the low-affinity receptor) is elevated, and due to its low affinity, it is activated by immature forms of neurotrophins. This serves as a trigger for cascades leading to synaptic weakening, microglial activation, and pathological pruning. Increasing the proportion of mature NGF relative to immature forms promotes the restoration of the normal synaptic function and physiological pruning [185].

Thus, TrkA is responsible for the survival and maintenance of sensory neurons, while p75NTR is associated with pathological pain. This is why nonselective NGF antagonists (monoclonal antibodies) caused side effects, including joint necrosis, as blocking sensitivity led to the overuse of damaged joints in clinical trials.

#### 8.1.2. Resistance

##### P-Glycoprotein–Mediated Resistance

The failures of clinical trials with neurotrophins may be associated with various types of resistance, including multidrug resistance (MDR). MDR is a form of resistance to pharmacotherapy predominantly mediated by P-glycoprotein [186]. The elevated expression and activity of P-gp at the BBB alter the pharmacokinetics of psychotropic drugs. Decreased P-gp expression and activity at the BBB occur with aging and have been observed in neurodegenerative diseases. Therefore, polymorphisms in this protein, as well as its hyperexpression and hyperactivation, influence the clinical outcomes of neurotrophin-based pharmaceuticals. Primarily, P-glycoprotein-mediated resistance can develop against small molecule therapies and other psychotropic drugs, as P-glycoprotein is a major obstacle for them at the BBB [187].

##### Immune-Mediated Resistance

Clinical trial failures may be linked to rapidly developing immunoresistance. NGF is expressed in immune cells such as T and B lymphocytes, dendritic cells, and monocytes/macrophages [188]. In inflamed tissues, the NGF level increases in parallel with that of pro-inflammatory cytokines [189].

Furthermore, cytokine levels depend on neurotrophin levels and vice versa, forming multiple self-regulating cytokine–neurotrophin feedback loops. Microglia and astrocytes are responsible for maintaining the physiological homeostasis. During inflammation, microglia are converted into an activated state, M1 microglia, and express increased levels of tumor necrosis factor TNFα, IL-1β, and IL-6 [190]. These pro-inflammatory cytokines and chemokines recruit additional T cells and macrophages to the nerves. In response, Tregs produce anti-inflammatory cytokines like IL-10 and TGF-β, which attract anti-inflammatory A2 astrocytes that express neurotrophic factors GDNF and BDNF. GDNF attenuates microglial activation and reduces TNFα and IL-1β synthesis, leading to a reduction in acute inflammation and promoting a transition to a chronic state followed by neurodegeneration. On the other hand, pro-inflammatory cytokines are necessary for the activation of immune response but disrupt BDNF signaling, leading to neuronal dysfunction and increased vulnerability to injury.

The exact mechanism of this neuroimmune crosstalk remains unknown but may be due to the overlapping expression patterns of neurotrophic factors in the immune and nervous systems [191].

Neurotrophins suppress their own production through a feedback mechanism. In chronic inflammation, these regulatory connections are disrupted: inflammation → increased production of immature neurotrophins → depletion of neurotrophins → degeneration.

Prolonged induction of neurotrophins by immune cells can lead to incomplete maturation of neurotrophins and, consequently, impaired function. This results in neuroinflammation and cytokine-dependent neurotoxicity [192]. Oxidative stress and free radicals produced by activated glial cells can directly damage both neurotrophins and their receptors. Activated microglia and astrocytes release increased amounts of proteases that degrade neurotrophins, leading to resistance due to accelerated breakdown. It is possible that cases of rapidly progressive joint degeneration following anti-NGF treatment when co-administered with NSAIDs are associated with this type of resistance [193]. Thus, treatment ineffectiveness may be caused by resistance and unresponsiveness to neurotrophins due to the loss of their receptors, receptor polymorphisms, or accelerated degradation.

##### Resistance Mutations

Acquired resistance often occurs due to a loss of epigenetic control, resulting from new mutations in the genes coding for neurotrophins and their receptors.

Oxidative stress and injury can cause hypermethylation or histone acetylation at the BDNF promoter, leading to promoter hypermethylation and, as a result, a decline in its synthesis. Conversely, aerobic physical exercise causes hypomethylation and an increase in BDNF. HDAC inhibitors (e.g., valproic acid) increase BDNF expression, as demonstrated in mice and cell cultures [194,195]. Epigenetic regulation via miRNAs, such as miR-7-5p, clearly correlates with BDNF mRNA levels in patients with bipolar II disorder [196]. Regular aerobic exercise also regulates BDNF production at the epigenetic level.

Understanding the mechanisms of epigenetic regulation opens new therapeutic targets: HDAC inhibitors (valproic acid, vorinostat) can increase neurotrophin expression and have already shown neuroprotective potential in animal and cellular models of neurodegeneration, potentially helping to reduce mutation-associated resistance. However, resistance to HDAC inhibitors themselves can also develop [197].

#### 8.1.3. Delivery

Neurotrophins do not cross the BBB; therefore, these proteins cannot be used for neuroprotection following intravenous administration, and it is not feasible to administer these molecules via intracerebral injection in human stroke. In addition, they have a short plasma half-life [198]. The route of administration is crucial, as the efficacy of pharmaceuticals delivered to the central nervous system via intracortical, intrahippocampal, intracerebroventricular, and intranasal routes is more pronounced and prolonged compared to oral or intravenous administration.

Intracerebroventricular (ICV) infusion has limited efficacy, since most of the drug injected into the ventricles is rapidly distributed into the peripheral circulation rather than diffuses into the brain parenchyma. In animal models, uneven distribution of neurotrophins leads to gliosis. This could be one of the reasons why clinical trials with ICV infusion of GDNF for the treatment of Parkinson’s disease were terminated [199].

Small molecules that cross the BBB share two key molecular characteristics: the molecular weight below a threshold of 400–500 Da, and lipid solubility [200]. While there are many strategies to improve the solubility of substances, the molecular weight remains a critical limiting factor.

In the field of protein-based drug delivery, novel approaches are being explored. For example, a neuroprotective peptide incorporating TrkB and Tat sequences has recently been developed, which is capable of crossing both the BBB and the plasma membrane [201].

The delivery challenge is also relevant to AAV-based gene therapy. Astrocytes, glial cells, and neurons from other systems may also begin to express neurotrophins, while the target neurons may remain without the gene. Often, the therapeutic effect is observed only in the immediate vicinity of the injection site, leaving most of the target area untreated.

#### 8.1.4. Disease-Stage-Related Limitations

It is essential to consider the stage of the neurodegenerative disease during which neurotrophin therapy is administered, whether it be in the context of acute or chronic inflammation. During acute inflammation, there is an increased production of neurotrophins [202]. In chronic inflammation, the interplay between cytokines and neurotrophins is disrupted, creating a vicious cycle that cannot be broken simply by administering the exogenous neurotrophins [203]. It appears crucial to first reduce the hyperinflammation, restore receptor sensitivity, and only then introduce neurotrophins. Perhaps they should be administered to areas adjacent to the lesion site, where the regulatory relationships between cytokine and neurotrophin production remain intact. Such an approach has already been demonstrated in treating vitiligo: under the influence of topical antioxidants, healthy melanocytes migrate into the affected area and proliferate [204].

The dosage and administration regimens of neurotrophins depend on the disease stage. Course-based neurotrophin therapy appears optimal for neurodegenerative diseases, whereas single high-dose administration may be preferable for TBI.

### 8.2. Future Prospects

Based on the aforementioned limitations of using neurotrophin-based pharmaceuticals, several key factors can be identified as critical for their development and subsequent clinical research.

High selectivity. It is advisable to prioritize the development of highly selective agonists and antagonists of neurotrophins capable of binding exclusively to a specific receptor subtype.Advanced delivery systems. To enhance the blood–brain barrier penetration, the development of novel delivery complexes, such as mesenchymal stem cell-derived extracellular vesicles [205], is recommended. For small molecules, the focus should be on creating new drug formulations such as nanocarriers or liposomes.Optimized administration and technique. Optimizing and standardizing the administration routes, surgical techniques, and injection procedures are recommended, integrating imaging guidance to simplify and reduce the cost of these interventions. For gene therapy, the use of new vectors with consideration for AAV serotype tropism is advised [206,207].Optimal dosing. The dosage of neurotrophins is a highly debated topic. Regarding BDNF, the data are controversial: some studies indicate that overdosing can cause pro-epileptogenic effects in animals and increases neuronal excitability [208], while others show that the overexpression of BDNF leads to the protection of brain cells from epileptic seizures [209,210].Personalized pharmacogenomics. Prior to clinical trials, creating the personalized pharmacogenomic reports for enrolled patients is recommended. This involves assessing the polymorphisms in neurotrophin and receptor genes, as well as in the detoxification system (e.g., P-gp, cytochrome P450 isoforms, GST isoforms), for instance, the Val66Met polymorphism in BDNF can negate synaptic plasticity and treatment outcomes [211].Capability to distinguish pro- and mature forms of NTs. Meticulous distinction between the pro- and mature forms of neurotrophins, such as pro-BDNF and mature BDNF, and the evaluation of their respective levels, are recommended [212].Assessing the disease status with respect to neuroinflammation. It is recommended to account for the extent of neuroinflammation using methods such as TSPO PET imaging, measuring sTREM2/GFAP (soluble triggering receptors expressed on myeloid cells-2/glial fibrillary acid protein) in CSF [213], plasma/CSF neurofilament light chain (NfL) level, and cytokine profiling. Monitoring the dynamics of these pharmacodynamic markers is crucial for assessing the direct therapeutic effect.Transition to plasma biomarkers. A shift from reliance on CSF markers to incorporating the validated plasma biomarkers is needed. Systemic inflammation markers, such as C-reactive protein, should consistently be included in the analysis.Account for comorbidities. Considering comorbid conditions, such as insulin resistance, is essential. It is also important that no other disease-modifying therapies be administered for at least two months prior to the start of the study period.

The endogenous neurosteroids P4, DHEA, and allopregnenolone in combination with aerobic physical exercise, as well as the use of biofeedback and adaptive stimulation can be good well-tolerated strategies for the neuroprotection [214], but more specific pharmaceuticals are needed. The development of recombinant NGFs and selective NGF receptor modulators without serious side effects is urgently needed. The approval of revakinagene taroretcel and cenegermin and approaching the approval of Levicept confirm good prospects of these pharmacological strategies for the treatment of neurodegenerative diseases and neuroinflammation.

## 9. Conclusions

The modulation of the expression and activity of neurotrophins, as well as that of their receptors, represents a promising strategy for treating various neurological disorders. Despite a substantial number of animal and human studies devoted to recombinant neurotrophins and the modulators of their receptors, only a few have received the approval of the FDA: Encelto (revakinagene taroretcel) and Oxervate (cenegermin). The peptide p75NTR antagonist Levi-04 which has shown positive Phase II results is currently in the final stages of investigation, with results eagerly awaited. LM11A-31, which is planning for a Phase III clinical trial for AD, along with GDNF gene therapy for neurological diseases, are among the most advanced therapeutic strategies.

This limited success is attributed to several challenges: neurotrophins and their agonists can induce side effects such as pain, provoke antibody development and immunogenicity, and often fail to penetrate specific brain regions. Conversely, neurotrophin antagonists can disrupt nociception to an extent that the resulting imbalance leads to increased tissue wear and necrosis, as observed in osteoarthritis models. Furthermore, genetic polymorphisms in neurotrophins and their receptors contribute to the heterogeneity of clinical trial outcomes. Finally, the long-term side effects, carcinogenicity, and mutagenicity of neurotrophin-based pharmaceuticals remain insufficiently investigated.

Neurotrophin-based pharmaceutics are breakthrough technologies in the treatment of neurodegenerative diseases and neuroinflammation. Many trials end in failure, but analyzing these failures helps develop new strategies. The introduction of recombinant NGF and CNTF for ocular diseases into clinical practice can partially explain the other failures. Access to the retina and the optic nerve is easier compared to other CNS structures, as there is no need to cross the BBB or pass through the systemic circulation. A crucial point is that the eye possesses the mechanisms of active immune suppression [215], which reduces the risk of a powerful immune-mediated inflammatory response to the administered neurotrophin. Retinal neurons and their axons represent discrete, compact cell populations, making the tropism of the neurotrophin delivery vector optimal in this context. We anticipate the approval of new therapies in the near future, since “Every success in medicine as a whole is reflected in the achievements of ophthalmology, and each success of the latter is reflected in the successes of medicine as a whole”—V.P. Filatov, the great ophthalmologist of the 20th century.

## Figures and Tables

**Figure 1 medsci-14-00015-f001:**
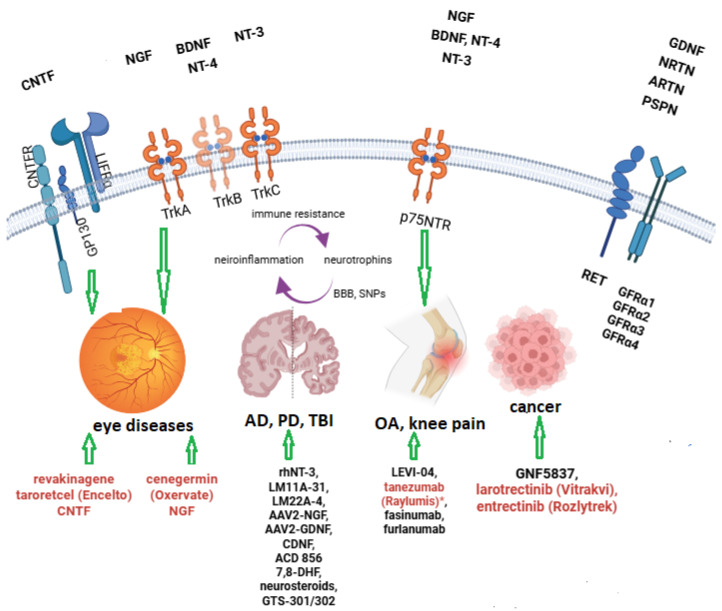
Neurotrophin-based pharmaceuticals for the treatment of neurodegenerative diseases and neuroinflammation. Note: Pharmaceuticals approved by the FDA are marked in red, and pharmaceuticals under development are shown in black. “*” is approved, but not yet introduced to the pharmaceutical market. All of them are agonists or antagonists of neurotrophin receptors. The arrows indicate the receptor-specific direction of action for the pharmaceutics. A closed loop of obstacles—permeability through the BBB, gene polymorphisms (single nucleotide polymorphisms, SNPs), and immune resistance—hinders the approval of BDNF and NT-3 mimetics. The figure was created using Biorender (https://biorender.com/, accessed on 12 December 2025).

**Table 1 medsci-14-00015-t001:** The main functions of neurotrophins.

NT	NGF	BDNF	NT-3	GDNF	CNTF
**Structure**	Belongs to the family of neurotrophins	Belongs to the family of neurotrophins	Belongs to the same family as NGF and BDNF	Distantly related to TGF-β family	IL-6 family member
**Receptor**	High-affinity receptor TRkA and low-affinity receptor LNGFR/p75NTR	TRkB and LNGFR/p75NTR	TrkC	GFRα1 and RET	gp130, LIFR, CNTFR
**Main function**	Inhibits neuronal apoptosis, promotes axon regeneration and angiogenesis through its interaction with TrkA, a high-affinity receptor; induces pain through its interaction with p75NTR [5,146]	Regulates synaptic plasticity [12], prevents microglial activation after hypoxic stimulation [147]	Maintains striatal synaptic plasticity [148] and neuronal differentiation [149]	Improves development and survival of dopaminergic neurons [106]	Activates CNTFR and the two signaling β-receptors glycoprotein 130 (gp130) and leukemia inhibitory factor receptor (LIFR). Neuroprotective cytokine [150]

**Table 2 medsci-14-00015-t002:** The main advances and the neurotrophin-based pharmaceuticals for the treatment of neurodegenerative diseases and neuroinflammation.

Drug	Mode of Action	Disease	Delivery Route	Key Results	Development Stage
**Agonists/mimetics of NGF**					
**Cenegermin** (Oxervate™) [27,150]	The recombinant human NGF	Neurotrophic keratitis	Eye drops	Restores corneal integrity	Approved
**hNGFp** [28]	The recombinant mutated form of human hNGF with reduced binding affinity to p75NTR	Retinitis pigmentosa	Eye drops	Reduces microglia-mediated inflammation	Investigational
**AAV2-NGF** [151,152]	The recombinant human NGF	AD	Intracerebral injections	No significant effect	Phase II clinical trial
**Antagonists of NFG**					
**Anti-NGF antibody** Tanezumab, fasinumab, furlanumab [43,44,45]	Anti-NGF antibody	OA	Intravenous injection, subcutaneous injection	Relieves pain but destroys the joint due to overload	Clinical trials stopped
**Anti-NGF antibody** DS002 [153]	Blocks NGF binding to TrkA	OA	Subcutaneous injection	Alleviates chemotherapy-induced peripheral neuropathy in rats	Investigational
**LEVI-04** [47]	p75NTR fusion protein designed to bind excess NGF	OA	Intravenous injection	Clinically meaningful improvement in pain, function, and other outcomes	Phase II clinical trial
**LM11A-31** [154]	BBB—penetrating small molecule, p75NTR antagonist	AD	Oral	Slows progression of pathophysiological features of AD	Phase II clinical trial
**PD90780** [155]	NGF antagonist, prevents it from binding to p75NTR	Potential anti-cancer drug	In vitro, i.p.	Experimental chemical compound for in vitro studies	Investigational
**Ro08-2750** [156]	Reversible NGF inhibitor	potential anti-cancer drug	i.p.	Inhibition of tumor growth (myeloid leukemia)	Investigational
**ASP7962** [157]	TrkA inhibitor	OA	Oral	Analgesic efficacy	Phase II clinical trial
**GZ 8998A** [158]	TrkA inhibitor	OA	Intra-articular	Analgesic efficacy	Phase II clinical trial stopped
**Agonists/mimetics of BDNF**					
**ACD856** [70,71]	Allosteric activator	AD	Intravenous and oral administration		Phase 1 clinical trial
Agomelatine [159] ketamine [160] ACTH-analogues [161] esketamine [162] neurosteroids DHEA, progesterone, Allo [72] cerebrolysin [163]	Indirect stimulation of BDNF synthesis via enhancement of TrkB phosphorylation	Different indications	Different routes	Neuroprotective action	Approved, but not for neurodegenerative diseases
**Cell and gene technologies** [164]	Amyloid beta-protein (Aβ)-specific CD4 T cells, genetically engineered to express BDNF	AD	Intracerebroventricularly	Reduced levels of beta-secretase 1 (BACE1)-a protease essential in the cleavage process of the amyloid precursor protein-and ameliorated amyloid pathology and inflammation within the brain parenchyma	Investigational
**AAV-BDNF** [75]	Gene–cell construct with an adenoviral vector encoding mature BDNF	AD	Intrathecally	Improved motor activity of the hind limbs and reduced the size of cysts in rats	Investigational
**Antagonists of BDNF**					
**K252a** [165]	Non-selective Trk inhibitor	TBI, tumors	Intracerebroventricular infusion	Prevents brain damage, reduces tumor growth	Investigational
**Ana12** [166]	Selective TrkB antagonist	TBI	intraperitoneal injection	Reduces pain behaviors and promotes locomotor function recovery	Investigational
**Larotrectinib** (LOXO-101) and **Entrectinhib** [167]	Trk kinase inhibitors	NTRK fusion-positive cancers	oral	Inhibits tumor cell growth and survival	Approved
**EG00229** [168]	Inhibitor of co-receptor NRP1 (neurolipin-1)	neuropathic pain	In vitro	Suppresses NGF-stimulated excitation of human and mouse nociceptors neurons in vitro; causes a concentration-dependent inhibition of NGF-induced sensitization of transient receptor potential vanilloid-1 (TRPV1) on nociceptors	Investigational
**Agonists/mimetics of NT-3**					
**rhNT-3** or TAT [169]	Human HIV-produced transactivator of transcription)-fused recombinant neurotrophin-3 T-NT-3 to enhance NT-3 delivery	AD	Intraperitoneal injection	Inhibits oxidative stress, apoptosis, and inflammatory responses in neural cells by activating TrkC receptors and suppressing M1 microglial activation. In vivo, T-NT-3 improves cognitive and memory impairments in mice	Investigational
**LM22B-10** [170]	TrkB/TrkC-activating compound	TBI, AD	Eye drops	Improves the healing speed of the corneal epithelium, corneal sensitivity, and corneal nerve density in regular and diabetic mice with corneal wounding	Investigational
**GTS-302** [171]	TrkB/TrkC activating dipeptide	-	Intraperitoneal	Exhibits antidepressant-like activity anxiolytic and memory-enhancing activity; does not affect pain sensitivity in mice	Investigational
**Agonists/mimetics of GDNF**					
**AAV2-GDNF** [115]	Gene therapy	PD	Bilateral intraputaminal delivery	Well tolerated and associated with numerical stability (mild cohort) and improvement (moderate cohort) in clinical assessments at 18 months posttreatment	Phase Ib clinical trial
**CNS10-NPC-GDNF** [116]	Human neural progenitor cells secreting GDNF	ALS	Unilateral, motor cortex	One administration of engineered neural progenitors can provide new support cells and GDNF delivery to the ALS patient spinal cord for up to 42 months post-transplantation	Phase I/IIa clinical trial
**Neublastin** (artemin, **BG00010**) [117,118,119,120,121]	Selective ligand for the GDNF family receptor alpha-3 (GFRα3) co-receptor	Neuropathic pain, sciatica	Intravenous or subcutaneous	Anti-hyperalgesic effects	Investigational
**CDNF** [123,172]	Recombinant human CDNF	PD	Intraputamenally	No significant changes in motor symptom assessment between placebo and CDNF treatment groups	Phase I clinical trial
**BT13** [125], **BT18** [173]	RET receptor agonists	Glaucoma, neuropathy	Subcutaneous injections	Reduces mechanical hypersensitivity and restores the normal expression levels of sensory neuron markers in dorsal root ganglia	Investigational
**Agonists/mimetics of CNTF**					
**Dapiclermin (Axokine)** [140]	**CNTF derivative**	ALS, obesity and insulin resistance	Intravenous injection	Average weight loss of 6 pounds compared to 2 pounds in patients given a placebo; 70 percent of patients developed blocking antibodies that limited its continued effectiveness	Approve
**NT-501, Revakinagene taroretcel (ENCELTO™)** [142,174]	**Recombinant CNTF**	Idiopathic MacTel type 2	Single-dose intravitreal implant	Induces a cascade of signaling events that promote photoreceptor survival	Approved

## Data Availability

No new data were created or analyzed in this study.

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
