# Peer review of "Current State of the Neurotrophin-Based Pharmaceutics in the Treatment of Neurodegenerative Diseases and Neuroinflammation"

_medsci, 2025, doi:10.3390/medsci14010015_

Round 1
Reviewer 1 Report
Comments and Suggestions for Authors
The manuscript addresses a well-known and extensively reviewed topic, namely the pharmacological exploitation of neurotrophins and their receptors in neurodegenerative diseases and neuroinflammation. While the topic is of undoubted biomedical relevance, the present review does not provide a clearly identifiable novel conceptual contribution when compared with multiple high-quality reviews published over the last 10–15 years in journals such as Trends in Neurosciences, Nature Reviews Drug Discovery, Progress in Neurobiology, Neurobiology of Disease, and Pharmacological Reviews.
The manuscript largely adopts a descriptive catalog-style approach, sequentially summarizing NGF, BDNF, NT-3, GDNF and CNTF pharmacology. However, the work lacks a unifying mechanistic framework, a strong critical perspective, and a forward-looking translational interpretation, all of which are currently expected for a review in a high-impact pharmacological journal.
In its present form, the manuscript mainly reproduces consolidated knowledge without offering a new interpretative model, a meta-analysis of failures vs. successes, or a systems-level vision linking neurotrophin signaling to disease heterogeneity and patient stratification. Therefore, the originality is limited.
A major weakness of the manuscript is the predominantly narrative and non-analytical structure. Although a large amount of literature is cited, the paper rarely provides true critical comparison between competing strategies, nor does it sufficiently interrogate:
- Why specific clinical trials failed beyond generic explanations (BBB penetration, immunogenicity).
- Whether failure was target-related, delivery-related, or disease-stage-related.
- Differences between receptor subtype activation vs. pan-neurotrophin stimulation.
- Disease-specific versus system-wide consequences of neurotrophin modulation.
For example:
- The discussion on NGF antagonists in osteoarthritis is thorough descriptively, yet lacks a mechanistic analysis of why TrkA versus p75NTR blockade leads to distinct safety profiles.
- BDNF signaling is treated largely as uniformly neuroprotective, without sufficient emphasis on its well-documented pro-epileptogenic and maladaptive plasticity roles, which are only briefly mentioned.
- GDNF gene therapy trials are summarized but not critically dissected in terms of vector tropism, distribution gradients, surgical variability, or neuroimmune activation.
Overall, the manuscript would benefit from a much stronger mechanistic selectivity and disease-contextual interpretation rather than a receptor-by-receptor pharmacological inventory.
Limitations
- Lack of a Guiding Hypothesis
The review does not formulate a clear central hypothesis (e.g., why neurotrophin-based therapies repeatedly fail in late-stage neurodegeneration but succeed in ocular diseases). - Insufficient Integration of Neuroinflammation
Although neuroinflammation is included in the title, its treatment remains superficial and fragmented, without:
- Dedicated sections on microglia–neurotrophin cross-talk,
- Cytokine–neurotrophin feedback loops,
- Immune-mediated resistance to trophic therapies.
- Absence of Precision Medicine Perspective
The manuscript mentions genetic polymorphisms but does not develop: - A stratification framework for responder vs. non-responder patients,
- A pharmacogenomic model for neurotrophin-based therapeutics.
- Minimal Translational Roadmap
The Conclusions reiterate known limitations without proposing: - Concrete optimization strategies,
- Biomarker-guided clinical trial designs,
- Combination approaches with neuroinflammation modulators, epigenetic drugs, or metabolic regulators.
The literature coverage is reasonably broad but skewed toward classical neurotrophin pharmacology, while several rapidly evolving areas are underrepresented or missing:
- Epigenetic regulation of neurotrophin expression.
- Neurotrophin control of synaptic pruning in neurodegeneration.
- Interactions with metabolic dysfunction.
- Sex-dependent neurotrophin signaling.
- Neurotrophin–tumor microenvironment interactions outside of generic NGF–cancer statements.
Furthermore, several citations are dated, and the manuscript would strongly benefit from the inclusion of high-impact 2020–2025 mechanistic reviews and clinical meta-analyses.
Figures and Tables
- Figure 1 is visually clear but overly schematic and not mechanistically informative.
- Table 1 is extensive but functions as an inventory rather than an analytical resource.
- No comparative or predictive modeling is presented (e.g., BBB penetration vs. molecular weight vs. clinical outcome).
Final Recommendation
Major Revision Required and Limited Originality in Current Form
While the manuscript is technically accurate and sufficiently referenced, its current contribution remains mainly archival rather than conceptual or forward-looking. To reach publishable standards at an international level, the authors must:
- Introduce a strong unifying conceptual framework.
- Deeply analyze the causes of clinical trial failures using mechanistic and translational criteria.
- Integrate neuroinflammation more rigorously at molecular and cellular levels.
- Adopt a precision-medicine and patient-stratification perspective.
- Strengthen the translational roadmap with concrete, testable therapeutic strategies.
Without these major revisions, the manuscript remains a competent but largely non-innovative overview of an already extensively reviewed field.
Author Response
Dear Reviewer,
We sincerely thank you for your thorough and deep review. Your comments are highly valuable and have allowed us to introduce a more critical perspective and strengthen the analytical aspect of the manuscript. The key sections of the article have been re-written and structured.
We have addressed all your suggestions and clarifications, although, in our view, each point merits its own dedicated review.
In the revised version, we have described the causes of failures and future opportunities, based on the points you highlighted. We have revised the abstract, conclusion, figure, and table to ensure the review's core concept is foregrounded—moving beyond a mere description of strategies to include their critique and future perspectives.
We have also added a new chapter on limitations and a roadmap for future research, as other reviewers expressed the same request. Thank you for your constructive suggestions—we have structured the chapter into subsections according to your recommendations. In this chapter, we have addressed every one of your comments, and 38 new references have been added.
Best regards,
The authors
Reviewer 2 Report
Comments and Suggestions for Authors
In this review, the authors examine and summarize the development of pharmaceuticals based on neurotrophins and neurotrophic factors for treating neurological and other disorders. While these molecules are critical for neuronal survival and plasticity, clinical success has been limited due to delivery challenges (such as the blood-brain barrier) and side effects like pain. To date, only a few associated drugs have received FDA approval, primarily for ocular diseases, while several others are still undergoing clinical trials for various diseases.
Suggestions:
--Please modify some sentences in the introduction section, such as: "Unfortunately, until now, most clinical trials involving recombinant nerve growth factors and various activators of their receptors have failed to show success in treating neurodegenerative diseases such as Alzheimer's disease (AD), Parkinson’s disease (PD), or multiple sclerosis." It is important to note that for some clinical trials, we cannot label them as “failures” if the FDA or the company has not made any announcements. This is a serious issue because many researchers are still working on those candidates.
--In the conclusion section, the authors should add more information, such as Encelto and Oxervate (both FDA-approved) for eye conditions; Levi-04, which is in clinical trials and has shown positive Phase II results for OS; and LM11A-31, which is planning for a Phase III clinical trial for AD, along with GDNF gene therapy for neurological diseases.
-- Most importantly, the authors should include Cerebrolysin in this review, as it has been approved in many countries and should not be missed.
--In Section 1, the authors should provide the search methods, timeline, and research areas for this comprehensive review.
--The author should also cite the first paper on all drugs, which is the most important contribution to the field.
Author Response
Dear Reviewer, thank you very much for your comments and suggestions.
- Please modify some sentences in the introduction section, such as: "Unfortunately, until now, most clinical trials involving recombinant nerve growth factors and various activators of their receptors have failed to show success in treating neurodegenerative diseases such as Alzheimer's disease (AD), Parkinson’s disease (PD), or multiple sclerosis." It is important to note that for some clinical trials, we cannot label them as “failures” if the FDA or the company has not made any announcements. This is a serious issue because many researchers are still working on those candidates.
Thank you for the suggestion. We have deleted the sentence: “There is no evidence for their high effectiveness in neuroprotection and neurorehabilitation”. We have rephrased the sentence as follows:
“Promising efficacy data for LM11A-31, now progressing to Phase III trials for Alzheimer’s disease, has been reported. For many neurotrophin-based pharmaceutics, favorable safety profiles in early‑phase studies support the potential for clinically meaningful efficacy in later‑stage trials”.
- In the conclusion section, the authors should add more information, such as Encelto and Oxervate (both FDA-approved) for eye conditions; Levi-04, which is in clinical trials and has shown positive Phase II results for OS; and LM11A-31, which is planning for a Phase III clinical trial for AD, along with GDNF gene therapy for neurological diseases.
We agree. We have strengthened the conclusions section as suggested. The primary additions are:
The modulation of the expression and activity of neurotrophins, as well as that of their receptors, represents a promising strategy for treating various neurological disorders. Despite a substantial number of animal and human studies devoted to recombinant neurotrophins and the modulators of their receptors, only a few have received the approval of the FDA: Encelto (revakinagene taroretcel) and Oxervate (cenegermin). The peptide p75NTR antagonist Levi-04 which has shown positive Phase II results is currently in the final stages of investigation, with results eagerly awaited. LM11A-31, which is planning for a Phase III clinical trial for AD, along with GDNF gene therapy for neurological diseases, are among the most advanced therapeutic strategies.
This limited success is attributed to several challenges: neurotrophins and their agonists can induce side effects such as pain, provoke antibody development and immunogenicity, and often fail to penetrate specific brain regions. Conversely, neurotrophin antagonists can disrupt nociception to an extent that the resulting imbalance leads to increased tissue wear and necrosis, as observed in osteoarthritis models. Furthermore, genetic polymorphisms in neurotrophins and their receptors contribute to the heterogeneity of clinical trial outcomes. Finally, the long-term side effects, carcinogenicity, and mutagenicity of neurotrophin-based pharmaceuticals remain insufficiently investigated.
Neurotrophin-based pharmaceutics are breakthrough technologies in the treatment of neurodegenerative diseases and neuroinflammation. Many trials end in failure, but analyzing these failures helps develop new strategies. The introduction of recombinant NGF and CNTF for ocular diseases into clinical practice can partially explain the other failures. Access to the retina and the optic nerve is easier compared to other CNS structures, as there is no need to cross the BBB or pass through the systemic circulation. A crucial point is that the eye possesses the mechanisms of active immune suppression [215], which reduces the risk of a powerful immune-mediated inflammatory response to the administered neurotrophin. Retinal neurons and their axons represent discrete, compact cell populations, making the tropism of the neurotrophin delivery vector optimal in this context. We anticipate the approval of new therapies in the near future, since “Every success in medicine as a whole is reflected in the achievements of ophthalmology, and each success of the latter is reflected in the successes of medicine as a whole” - V.P. Filatov, the great ophthalmologist of the 20th century.
- — Most importantly, the authors should include Cerebrolysin in this review, as it has been approved in many countries and should not be missed.
We agree. We have added Cerebrolysin to the table as an indirect mimetic and provided a reference. Cerebrolysin has been shown to increase BDNF levels, as recently demonstrated. Since this preparation is a mixture of peptides derived from pigs, it is difficult to pinpoint which specific peptide is responsible for the clinically significant effect.
- — In Section 1, the authors should provide the search methods, timeline, and research areas for this comprehensive review
We agree. Our review is more narrative than comprehensive. Currently, there exists no published work that both classifies neurotrophins and reviews the current landscape of drugs in development aimed at modulating their activity. We have included a Methods section in the abstract outlining the criteria used for literature selection and analysis.
- — The author should also cite the first paper on all drugs, which is the most important contribution to the field.
Thank you. We agree. We have reorganized the references in the table, aiming to select the specific publication that demonstrates an effect on a particular neurotrophic factor—whether agonistic or antagonistic—as this was the core concept of the review.
Reviewer 3 Report
Comments and Suggestions for Authors
This manuscript has significant strengths that support publication with revisions:
- Comprehensive coverage of multiple neurotrophin families and their receptors
- Excellent inclusion of recent developments including 2024-2025 approvals and clinical trials
- Good integration of molecular mechanisms with clinical applications
- Balanced presentation of both agonist and antagonist therapeutic strategies
- Recognition of dual roles of neurotrophins (e.g., BDNF in acute vs chronic inflammation, GDNF in neuroprotection vs hyperdopaminergia)
- Extensive, up-to-date reference list (>170 references)
There are several groups of issues that have to be addressed before publication:
1. The review methodology is not systematic:
- No clear search strategy is provided (databases searched, search terms, date ranges)
- Inclusion/exclusion criteria for studies are not defined
- The selection process for clinical trials discussed is unclear
- No quality assessment of included studies is performed
You should add a Methods section, detailing the literature search strategy... or explicitly state it is a narrative review.
The critical analysis is lacking, because, while trials are mentioned, it is rarely discussed why there have been failures, why gene therapy approaches were more successful than protein therapies, also the controversial results of NGF. A paragraph should be added, analyzing patterns of success and failure across trials.
In the classification section, lines 65-118, systems are presented without any justification of utility. Information is poorly organized between trials, basic biology, pharmacology and clinic without clear limitations. Use a consistent structure for each neurotrophin.
Also, the delivery systems seem insufficiently discussed. A paragraph or section should be added comparing delivery methods, with advantages, problems, current status, passage through the BBB and so on.
Among the recommendations:
- Completely redesign the summary table: there are several tables (probably a formatting issue) and there is no number mention. Consolidate them into one comprehensive table with columns for: Drug Name, Neurotrophin Target, Mechanism, Disease Indication, Delivery Route, Development Stage, Key Results, Current Status
- Add a Methods section or clearly state this is a narrative review with appropriate scope limitations
- Expand critical analysis: add a dedicated section analyzing patterns of trial success/failure with specific discussion of trial design, patient selection, outcome measures, and dosing strategies
- Strengthen the conclusions section to synthesize key findings and provide specific recommendations for future research
Comments on the Quality of English Language
The manuscript is generally well-written but would benefit from English language editing:
- Inconsistent hyphenation and spelling throughout
- Some sentences are overly long and complex, reducing clarity
- Citation formatting is inconsistent
Author Response
Dear Reviewer, thank you very much for your comments and suggestions.
- The review methodology is not systematic. You should add a Methods section, detailing the literature search strategy... or explicitly state it is a narrative review.
We agree, we have included a Methods section in the abstract outlining the criteria used for literature selection and analysis.
- The critical analysis is lacking, because, while trials are mentioned, it is rarely discussed why there have been failures, why gene therapy approaches were more successful than protein therapies, also the controversial results of NGF. A paragraph should be added, analyzing patterns of success and failure across trials.
We agree, we have added a dedicated critical discussion chapter “ 7. Limitations and future prospects for the neurotrophin-based pharmaceutics” and systematically restructured the table.
- In the classification section, lines 65-118, systems are presented without any justification of utility. Information is poorly organized between trials, basic biology, pharmacology and clinic without clear limitations. Use a consistent structure for each neurotrophin. Also, the delivery systems seem insufficiently discussed. A paragraph or section should be added comparing delivery methods, with advantages, problems, current status, passage through the BBB and so on.
We agree, (lines 65-118). We have added the following justification of utility in the classification section: “Currently, there exists no published work that classifies neurotrophins. The unified classification of neurotrophins will aid in optimizing neurotrophin-targeted strategies for treating neurodegenerative and neuroinflammatory diseases».
We agree, while an analysis of the reasons behind clinical trial failures could be the subject of a separate review, our goal here was to illustrate which neurotrophin modulation strategies, in principle, hold therapeutic potential for medical application.
To highlight the key points and clearly illustrate the current state of research, we have revised the table and added a dedicated concluding chapter 7. The table has been restructured to include new columns for evaluating the current development status of neurotrophin agonists and antagonists. We have completely redesigned the summary table.
- Add a Methods section or clearly state this is a narrative review with appropriate scope limitations
We agree. We have included a Methods section in the abstract outlining the criteria used for literature selection and analysis.
- Expand critical analysis: add a dedicated section analyzing patterns of trial success/failure with specific discussion of trial design, patient selection, outcome measures, and dosing strategies.
We appreciate this insightful suggestion and fully agree. While our review initially catalogued agonists and antagonists of neurotrophins as potential strategies against neurodegeneration and neuroinflammation, we have now incorporated a dedicated critical analysis section «Limitations and future prospects for the neurotrophin-based pharmaceutics». This new chapter systematically examines the limitations and translational challenges of these approaches, with a focused discussion on trial design, patient stratification, and outcome measures. To conclude, it provides a forward-looking roadmap, distilling our analysis to highlight the most promising therapeutic avenues for future development.
- Strengthen the conclusions section to synthesize key findings and provide specific recommendations for future research
We agree, we have strengthened the conclusions section as suggested. The primary additions are:
The modulation of the expression and activity of neurotrophins, as well as that of their receptors, represents a promising strategy for treating various neurological disorders. Despite a substantial number of animal and human studies devoted to recombinant neurotrophins and the modulators of their receptors, only a few have received the approval of the FDA: Encelto (revakinagene taroretcel) and Oxervate (cenegermin). The peptide p75NTR antagonist Levi-04 which has shown positive Phase II results is currently in the final stages of investigation, with results eagerly awaited. LM11A-31, which is planning for a Phase III clinical trial for AD, along with GDNF gene therapy for neurological diseases, are among the most advanced therapeutic strategies.
This limited success is attributed to several challenges: neurotrophins and their agonists can induce side effects such as pain, provoke antibody development and immunogenicity, and often fail to penetrate specific brain regions. Conversely, neurotrophin antagonists can disrupt nociception to an extent that the resulting imbalance leads to increased tissue wear and necrosis, as observed in osteoarthritis models. Furthermore, genetic polymorphisms in neurotrophins and their receptors contribute to the heterogeneity of clinical trial outcomes. Finally, the long-term side effects, carcinogenicity, and mutagenicity of neurotrophin-based pharmaceuticals remain insufficiently investigated.
Neurotrophin-based pharmaceutics are breakthrough technologies in the treatment of neurodegenerative diseases and neuroinflammation. Many trials end in failure, but analyzing these failures helps develop new strategies. The introduction of recombinant NGF and CNTF for ocular diseases into clinical practice can partially explain the other failures. Access to the retina and the optic nerve is easier compared to other CNS structures, as there is no need to cross the BBB or pass through the systemic circulation. A crucial point is that the eye possesses the mechanisms of active immune suppression [215], which reduces the risk of a powerful immune-mediated inflammatory response to the administered neurotrophin. Retinal neurons and their axons represent discrete, compact cell populations, making the tropism of the neurotrophin delivery vector optimal in this context. We anticipate the approval of new therapies in the near future, since “Every success in medicine as a whole is reflected in the achievements of ophthalmology, and each success of the latter is reflected in the successes of medicine as a whole” - V.P. Filatov, the great ophthalmologist of the 20th century.